# Physical Activity Dynamically Regulates the Hippocampal Proteome along the Dorso-Ventral Axis

**DOI:** 10.3390/ijms21103501

**Published:** 2020-05-15

**Authors:** Surina Frey, Rico Schieweck, Ignasi Forné, Axel Imhof, Tobias Straub, Bastian Popper, Michael A. Kiebler

**Affiliations:** 1Department for Cell Biology & Anatomy, Biomedical Center (BMC), Medical Faculty, Ludwig-Maximilians- University, 82152 Planegg-Martinsried, Germany; surina@lassfrey.de (S.F.); rico.schieweck@med.uni-muenchen.de (R.S.); 2Department for Molecular Biology (protein analysis unit), Biomedical Center (BMC), Ludwig-Maximilians-University, 82152 Planegg-Martinsried, Germany; ignasi.forne@lrz.uni-muenchen.de (I.F.); imhof@lmu.de (A.I.); 3Department for Molecular Biology (Core facility bioinformatics), Biomedical Center (BMC), Ludwig-Maximilians-University, 82152 Planegg-Martinsried, Germany; tstraub@bmc.med.lmu.de; 4Core Facility Animal Models, Biomedical Center (BMC), Medical Faculty, Ludwig-Maximilians-University, 82152 Munich, Germany; 5Institute of Pathology, School of Medicine, Technical University of Munich, 81675 Munich, Germany

**Keywords:** dorsal and ventral hippocampus, enhanced physical activity, mass spectrometry, protein expression, metabolism

## Abstract

The hippocampus is central for higher cognition and emotions. In patients suffering from neuropsychiatric or neurodegenerative diseases, hippocampal signaling is altered causing cognitive defects. Thus, therapeutic approaches aim at improving cognition by targeting the hippocampus. Enhanced physical activity (EPA) improves cognition in rodents and humans. A systematic screen, however, for expression changes in the hippocampus along the dorso-ventral axis is missing, which is a prerequisite for understanding molecular mechanisms. Here, we exploited label free mass spectrometry to detect proteomic changes in the hippocampus of male mice upon voluntary wheel running. To identify regional differences, we examined dorsal and ventral CA1, CA3 and dentate gyrus hippocampal subregions. We found metabolic enzymes and actin binding proteins, such as RhoA, being upregulated in the hippocampus upon EPA suggesting a coordination between metabolism and cytoskeleton remodeling; two pathways essential for synaptic plasticity. Strikingly, dorsal and ventral hippocampal subregions respond differentially to EPA. Together, our results provide new insight into proteomic adaptations driven by physical activity in mice. In addition, our results suggest that dorsal and ventral hippocampus, as well as hippocampal subregions themselves, contribute differently to this process. Our study therefore provides an important resource for studying hippocampal subregion diversity in response to EPA.

## 1. Introduction

Research in the last decade has unraveled the hippocampus as a central structure for learning and memory formation. Moreover, as part of the Papez circuit, the hippocampus is crucial for controlling emotion [1]. According to lesion experiments, spatial memory depends on the dorsal hippocampus (DH) [2] while stress response and emotional behavior rely on the ventral hippocampus (VH) [3]. This functional segregation is mirrored by distinct efferent and afferent pathways of the DH and VH, respectively [4]. Further, gene expression analysis revealed the existence of hippocampal subdomains along the longitudinal axis [5,6]. Together, these studies support the notion that a dorso-ventral hippocampal gradient exists that controls memory formation and emotional behavior. Thus, the anatomical architecture of the hippocampus allows the brain to link memory with emotions [1]. Importantly, the flexibility of both processes is crucially dependent on neurogenesis and the plasticity of established neuronal circuits [7]. Thereby, neuronal stem cells, residing in the dentate gyrus (DG) of the hippocampus, generate new neurons throughout life [8,9]. These neurons integrate into existing, mature brain circuits [10], a process that enables cognitive flexibility [7,11]. Together, the enormous plasticity of the hippocampus establishes this region as a central regulatory hub for higher cognition [8]. Hence, therapeutic strategies to treat, e.g., neurodegenerative diseases target the hippocampus to improve cognitive functions. One of these approaches is enhanced physical activity (EPA). EPA improves cognitive functions in both rodents [11,12,13] and humans at all ages [14,15,16,17]. Different mechanisms and molecular pathways have been found to play a role in EPA enhanced cognition including cardiovascular and immunological effects [17]. Interestingly, some of these pathways involve signaling via the insulin-like growth factor 1 (IGF-1) and the brain-derived neurotrophic factor (BDNF) [17]. Importantly, IGF-1 and BDNF are known to enhance neurogenesis in the DG [18,19] providing a physiological link between physical activity of the organism and the cellular consequences in the brain that lead to cognitive improvements. These findings strongly point towards a pivotal role of physical activity in cognition and provide novel approaches for therapies. Therefore, profiling the expression changes in the hippocampus is a prerequisite to understand the underlying pathways that are regulated by EPA.

Despite pioneer studies that aimed at unraveling the impact of EPA on hippocampus functioning, the molecular signature of this process is widely unknown. Therefore, we performed label-free quantitative mass spectrometry of hippocampi that came from either naïve or running wheel exposed mice. As the spatial architecture along the longitudinal axis contributes to synaptic plasticity, we decided to profile proteomic alterations in response to EPA in both DH and VH. Since hippocampal synaptic signaling relies on a complex interaction between distinct hippocampal subregions [8], we dissected proteomic changes in CA1, CA3 and DG. In detail, we identified metabolic enzymes to be upregulated in the hippocampus. Furthermore, we show that proteins involved in neuronal migration are dynamically altered. Strikingly, we observed stark differences between individual subregions within the hippocampus as well as between its dorsal and ventral parts. Our findings indicate that different molecular pathways become activated in response to physical activity. Based on our observations, we propose that the molecular signature of distinct hippocampal subregions selectively contribute to improvement in cognition upon EPA. Together, this is likely to provide the basis to develop therapeutic approaches for neuropsychiatric and neurodegenerative diseases.

## 2. Results

### 2.1. EPA Differently Impacts Adult Neurogenesis in Dorsal and Ventral Hippocampus

Running wheel exposure is a generally accepted approach to enhance neurogenesis in the DG [12,13]. To evaluate the impact of EPA on the dorsal and ventral hippocampal neurogenesis in the dentate gyrus (DG), we performed immunohistochemistry against doublecortin (DCX), a marker for immature neurons in the DG of the hippocampus [20,21] (Figure 1A). For all experiments, we used 10 week old mice to avoid neurogenesis-induced instability of memories as shown for infantile animals [11] that might have an opposing effect on cognitive improvement. Animals were voluntarily exposed to a running wheel for 3 weeks to allow the integration of newly born neurons into mature, established neuronal circuits [10]. For all experiments, we compared 10 week old, naïve animals with 13 week old, running wheel exposed, mice for the following reasons: (i) our strategy mimicked clinical conditions in which post- and pretreated conditions are compared. Particularly, physical activity has been suggested as a potential therapy to treat neurodegenerative diseases [17]; (ii) recent studies addressing proteomic changes during aging comparing 1 month or 6 months old rodents with 12 or 24 month old animals, respectively, reveal minor changes in the proteome (3%–5%) [22,23] indicating that aging is not significantly altering the steady-state proteome of the mature hippocampus. Supportive of this notion is a recent study showing that protein expression of 5 week and 20 week old animals is very similar [24]. Based on these results, we concluded that proteomic changes between 13 and 10-week-old animals with running wheel exposure are minor. We analyzed the number of DCX positive (DCX^+^) cells and surveyed the number of primary and secondary dendrites (Figure 1A,B,E). Interestingly, we observed a significant increase in the number of DCX^+^ cells in the ventral hippocampus, as also previously reported [11]. The effect in the dorsal part, however, did not reach statistical significance (Figure 1B,E). We also did not find a significant increase in the number of primary and secondary dendrites, respectively, for the dorsal hippocampus (Figure 1B,C). In contrast, ventral DCX^+^ cells revealed a significantly elevated number of primary and secondary dendrites (Figure 1E,F). Of note, we observed a higher variance in the number of DCX^+^ cells and number of primary as well as secondary dendrites in the naïve dorsal hippocampus. These results suggest a selective contribution of the dorsal and ventral hippocampus, respectively, to running wheel induced enhanced neurogenesis. The only parameter that was significantly increased in both parts of the hippocampus was the mean length of primary and secondary dendrites (Figure 1D,G). Importantly, we did not detect differences in number of DCX^+^ cells comparing 10 versus 13-week-old mice, respectively, without running wheel exposure (Appendix A). These results suggest that dorsal and ventral DG respond differentially to EPA. Therefore, we aimed at identifying the molecular signature accompanied with the running wheel exposure and, eventually, enhanced neurogenesis in the dorsal and ventral hippocampus.

### 2.2. Label Free Quantitative Mass Spectrometry Detects Proteomic Alterations during Neurogenesis

To analyze proteomic changes in response to running wheel exposure, we exploited label free, quantitative mass spectrometry of the dorsal and ventral DG, respectively. In addition, we also included the hippocampal subregions CA1 and CA3 to survey for their proteome dynamics in the process of neuron differentiation and integration, as they are part of the functional synaptic circuit within the hippocampus [8]. Therefore, we systematically sampled dorsal and ventral hippocampal subregions and analyzed protein expression by liquid chromatography coupled tandem mass spectrometry (LC–MS/MS; see Methods; Figure 2A).

First, we validated the reproducibility of our approach by comparing the detected proteins from dorsal and ventral CA1, CA3 as well as DG for five mice per group, respectively (Figure 2B). On average, we found 2244.6 ± 179.7 commonly detected proteins for all subregions from dorsal and ventral hippocampus (Figure 2B). Based on our analysis, we concluded that label free mass spectrometry would allow us to reproducibly detect proteomic changes in response to running wheel stimulation.

To profile first hippocampus-wide changes in protein expression, we merged all subregions from dorsal and ventral subregions and compared running wheel exposed with naïve mice (Figure 3A). Here, we identified a total of 70 proteins regulated upon running wheel stimulation. Strikingly, among those, 19 were described to be involved in neurogenesis, growth or apoptotic processes (Figure 3A). Amongst those proteins, we found Ras homolog family member A (RhoA) to be upregulated. RhoA is involved in actin stress fiber formation [25]. In line with this finding, the actin binding protein Profilin-2 (Pfn2) was also upregulated. Both proteins are involved in actin cytoskeleton remodeling [26], which represents an important intracellular process in synaptic plasticity [27]. Interestingly, Neuronal pentraxin-1 (Nptx1) also exhibited increased expression levels. It has been shown that Nptx1 regulates neuronal lineage specification [28]. Furthermore, we detected 35 upregulated proteins involved in metabolic pathways (Figure 3B and Appendix A). Metabolic enzymes such as mitochondrial succinate-semialdehyde dehydrogenase (SSDH), glucose-6 phosphate isomerase (G6PI), cytoplasmic aspartate aminotransferase (AATC), 2-oxoglutarate dehydrogenase (ODO1) and hypoxanthine-guanine phosphoribosyltransferase (HPRT) revealed elevated expression levels. Moreover, proteins involved in oxidative phosphorylation such as ATP-synthase-coupling factor 6 (Atp5j) or Cytochrome b-c1 complex subunit 1 (Uqcrc1) were significantly upregulated (Figure 3B). These findings probably reflect the high energetic costs of physical activity and, eventually, neurogenesis. Importantly, we also identified a possible link towards neurodegenerative diseases (Figure 3B). The involvement of neurogenesis as well as EPA in the pathology of neurodegenerative diseases such as Alzheimer’s, Parkinson’s and Huntington’s disease has been discussed [29,30,31,32].

### 2.3. Dorsal and Ventral Hippocampal Subregions Respond Differently to EPA

Synaptic signaling in the hippocampus relies on a complex interaction between the subregions CA1, CA3 and DG [8]. Thus, enhanced neurogenesis and integration of newly born neurons in DG might affect CA1 and CA3 through alterations in synaptic signaling. Therefore, we addressed proteomic changes in these subregions along the longitudinal axis. To survey for proteomic alterations with hippocampal subregion resolution, we imputed our mass spectrometry dataset as described previously [33]. This estimation allowed us to profile CA1, CA3 and DG separately (Figure 4A,B). Interestingly, we observed substantial and highly significant changes in the proteome of dorsal and ventral CA1, CA3 and DG. In addition, dorsal hippocampal subregions showed a higher number of affected proteins than ventral regions (compare Figure 4A,B, see also Figure 5A). These findings indicate different roles of dorsal and ventral CA1, CA3 and DG in coping with physical activity by running wheel exposure as well as in neurogenesis. To get further insight into the proteomic dynamics of this process, we compared the fold change of all proteins detected in the dorsal versus ventral DG caused by running wheel exposure. We observed that enhancing neurogenesis increases the proteomic diversity between dorsal and ventral DG (red dots in Figure 4C). These findings prompted us to speculate that EPA elevates differences in expression profiles between the hippocampal subregions along the dorso-ventral axis. This process might also account for differences in the number of DCX^+^ cells in the DG of DH and VH (Figure 1B,E). Indeed, comparing the proteomic changes between CA1, CA3 and DG from dorsal and ventral hippocampus, respectively, revealed that in all subregions distinct proteins were affected (Figure 5B). Therefore, it is tempting to speculate that DG, CA1 and CA3 exploit different expression programs to adapt to EPA. Accordingly, we observed a selective overlap for up- and downregulated proteins comparing dorsal and ventral subregions (Figure 5C) indicating that along the dorso-ventral axis cells respond differentially to voluntary exercise and, eventually, integration of newly born neurons. In line with these findings is our observation that different pathways were regulated differently in DH and VH, respectively (Appendix A). Strikingly, proteins involved in neuronal projections were found to be regulated in the DH, while proteins associated with metabolism and mitochondria exhibited expression alterations in the VH. Together, these observations support the notion that the DH and VH convey distinct aspects of EPA-induced proteomic alterations and therefore indicate functions for cognition.

In conclusion, our work provided first insight into the dynamics of the proteome within ventral and dorsal hippocampal subregions in response to EPA. We observed striking differences between hippocampal subregions as well as along the dorso-ventral axis. Our data indicate that physical activity drives the expression differences along the dorso-ventral axis.

## 3. Discussion

Research in the last decades has shown that EPA clearly affects cognition. One possible explanation for this effect is the promotion of neurogenesis in the hippocampus [17]. Importantly, the hippocampus is a very heterogeneous anatomical structure with great cell type diversity and functionally distinct ventral and dorsal parts [1]. Our data suggest that dorsal and ventral hippocampus responded differently upon EPA dependent neurogenesis. This finding is in line with two independent studies suggesting a higher neurogenesis efficiency in the ventral hippocampus [34,35] supporting functional segregation [7]. Importantly, lesion experiments have shown that the ventral hippocampus conveys emotional behavior while the dorsal hippocampus is essential for spatial learning [7]. Thus, distinct molecular pathways might convey the generation of neurons in these two structures. Moreover, several studies have shown that there is subregion specific transcriptional differences along the longitudinal axis [5,6,36]. Importantly, most of these studies exploit RNA sequencing approaches to investigate protein expression. However, protein expression weakly correlates with RNA levels [37] or ribosomal occupancy [38] for complex tissues. Thus, mass spectrometry approaches are needed to unravel proteomic changes in these tissues.

Therefore, our data suggest that hippocampal subregions display significant differences within their proteomes along the longitudinal axis. These differences might convey distinct cellular responses during EPA. Interestingly, the effect of physical activity on brain homeostasis might be highly selective. A pioneer study has shown that physical activity promotes angiogenesis in the cerebellum, while motor learning induces synaptogenesis [39]. Importantly, even though the ventral hippocampus seems to be more efficient in generating neurons, the dorsal hippocampus exhibits a more dynamic proteome. These effects might point towards feedback loops and cross-talks between the ventral and dorsal parts. In line with this notion is the observation that EPA promoted expression differences between DH and VH. Thus, it might be that increased differences in the molecular signature of DH and VH are a prerequisite for cellular adaptions during EPA. Future studies are clearly needed to unravel the molecular details of this process. Based on our findings, it is tempting to speculate that the interaction between dorsal and ventral hippocampus conveys cognitive flexibility that is needed for context learning but also for emotional behavior [7].

The beneficial effect of EPA on patients suffering from depression has been discussed [40]. Furthermore, depression leads to reduced neurogenesis [40]. Moreover, EPA might represent an alternative approach to treat patients suffering from neurodegenerative diseases [17]. Together, these studies strongly suggest a great therapeutic potential of physical activity to enhance higher cognition in health and disease. We think that our study might provide molecular insights into this process and how it shapes the hippocampal proteome. Moreover, we envisioned that our work would help to design and plan future studies.

Together, we concluded that dorsal and ventral hippocampus selectively adapted to EPA. We proposed that the balance between ventral neurogenesis and dorsal protein dynamics is a prerequisite for hippocampal homeostasis and might be an explanation for neuropsychiatric disorders.

## 4. Materials and Methods 

### 4.1. Animals

For all experiments, C57Bl/6JRj wild-type male mice (Janvier laboratories, Le Genest-Saint-Isle, France) aged 10 weeks or 13 weeks were used. All mice were kept under specified pathogen-free conditions and housed in groups of 2–3 animals in filter top cages and a 12 h/12 h light/dark cycle. Mice had free access to water (acidified and desalinated) and standard rodent chow (Altromin, 1310M). We used 10 animals (sacrificed with 10 weeks) for control and test group each. Test group animals were exposed to the running wheel for 3 weeks (sacrificed with 13 weeks). Home cages were equipped by a wireless, low-profile running table for voluntary use (Product ENV-047, Med Associates, St Albans, VT, USA). Running wheel activity was recorded during 24 h over 3 consecutive weeks. All experiments were approved by the authors’ institutional committee on animal care and were performed according to the German Animal Protection Law, conforming to international guidelines on the ethical use of animals (Az. 5.1-5682/LMU/BMC/CAM 2019-0007, approved on 02-12-2017).

### 4.2. Immunohistochemistry

#### 4.2.1. Tissue Preparation

For immunohistochemistry, tissue was prepared as previously described [41,42]. In brief, mice were deeply anaesthetized with CO_2_. Mice were transcardially perfused with PBS (pH 7.4) followed by 4% PFA (pH 7; Roti^®^-Histofix, Karlsruhe, Germany). Brains were post-fixed in 4% PFA (pH 7; Roti^®^-Histofix) for 12–72 h at 4 °C, then dehydrated in 30% sucrose in ddH_2_O at 4 °C for 24–48 h. Brains were cut into 40 µm coronal sections using a Leica cryotome (Product CM1850, Leica Microsystems, Wetzlar, Germany). Immunostaining was used as previously reported [41]. In brief, free-floating coronal brain sections were washed 3 times in PBS (pH 7.4), then blocked in blocking solution (1% (*w*/*v*) BSA, 0.5% (*v*/*v*) Triton X-100 in PBS) and incubated with primary antibodies overnight at 4 °C. Antibodies were diluted in blocking solution (rabbit anti-Doublecortin. DCX, 1:1000; Abcam); chicken anti-NeuN (1:500; Millipore, Darmstadt, Germany). Upon incubation, sections were washed in PBS and incubated with secondary antibodies (donkey anti-rabbit IgG Alexa Fluor 555, AF555, labeled and goat anti-chicken IgY Alexa Flour 647, AF647, labeled (both Life Technologies, Carlsbad, CA, USA) diluted 1:500 in blocking solution for 2 h. Upon washing, sections were mounted on Superfrost glass slides (Carl Roth, Karlsruhe, Germany) with Fluoromount (Sigma, Darmstadt, Germany).

#### 4.2.2. Tissue Selection

For systematic sampling of dorsal and ventral hippocampus, respectively, Bregma region −2.48 mm was used to anatomically distinguish these parts. We decided to sample along the dorso-ventral axis as learning and anxiety are differentially regulated along this axis [43]. For all mice, staining and imaging were performed on the same dorsally and ventrally orientated dentate gyrus to compare neuronal development between the control and test groups.

#### 4.2.3. Imaging

Confocal microscopy was performed with an inverted Leica SP8 microscope (Leica Microsystems, Germany) equipped with lasers for 405, 488, 552 and 638 nm excitation [41]. Images for quantification were acquired with a 40× 1.3 oil objective; overview images with a 20× 1.3 oil objective and the following fluorescence settings were used for detection: DAPI: 430–470 nm; AF555: 560–600 and AF647: 650–700. AF555 and AF647 were recorded with hybrid photo detectors and DAPI with a photomultiplier tube.

For each DG, we took the same number of tile scans (dorsal hippocampus: 6 tiles, ventral hippocampus: 7 tiles with a size per tile scan of 290.71 µm × 290.71 µm and 9.98 µm depth). Further, we analyzed entire layers of each dentate gyrus, respectively, the subgranular zone, the granule cell and the molecular layer. We analyzed one dorsal and one ventral DG per mouse per group (5 mice per control and test group, respectively).

#### 4.2.4. Quantification

Number of DCX^+^ cells as well as the number of primary and secondary dendrites were manually counted for each DG. Primary and secondary dendrites were distinguished according to their first branch point (Figure 1A). All DCX^+^ cell bodies localized in the subgranular and granular zone were included into the analysis.

### 4.3. Proteomics

#### 4.3.1. Tissue Preparation and Microdissection

Mice were deeply anaesthetized with CO_2_ and brains were quickly removed, transferred on dry ice and stored at −80 °C. Brains were dissected in 40 µm on-slide slices using a cryotome (Product CM1850, Leica Microsystems, Germany) as previously described [44]. Sections were subdivided according to the anatomical Bregma region −2.48 mm in a dorsal and a ventral part. Slides were stored at −80 °C. For subregion microdissection, slices were Nissl stained [41] and CA1, CA3 and DG separately removed under a stereo microscope (Wild, Gais, Switzerland).

The hippocampal neuropil and cell body layers were carefully microdissected from each slice (Figure 2A) using a stereomicroscope. For the DG, transversal cuts were made at the border of stratum lacunosum-moleculare and stratum moleculare of the DG. For the CA, cuts were made at the border of stratum lacunosum-moleculare and stratum oriens (Figure 2A). Lateral cuts were made at the end of the dentate gyrus axis, CA3-CA2, CA2-CA1 borders and near the end of region inferior in CA1. The CA2 region was excluded from sampling. To prepare sufficient tissue, we dissected both hippocampi from each brain of 5 wild-type mice per group, yielding 10 hippocampi and 200 microdissected slices. After microdissection, the tissue was transferred to a frozen tube containing 80% ethanol and stored at −80 °C prior to usage.

#### 4.3.2. Mass Spectrometry Analysis

Tissue preparation for mass spectrometry was performed using the Sample Preparation Kit (PREOMICS, Martinsried, Germany) following manufactures manual. For mass spectrometry, the peptide mixture resulting from the tryptic cleavage was injected onto an Ultimate 3000 RSLC HPLC system (ThermoFisher, Waltham, MA, USA) equipped with an analytical column (12 cm × 75 cm) home-packed with C18RP Reposil-Pur AQ (2.4 µm, 120 Å, Maisch, Germany) into an ESI-emitter tip (New Objective, Woburn, MA, USA). For peptide separation, a linear gradient from 5% to 40% B (HPLC solvents A: 0.1% FA, B:80% CAN, 0.1% FA) was applied over a time of 50 min. The HPLC was online coupled to an QExactive HF mass spectrometer (Thermo Fisher, Waltham, MA, USA). Data was analyzed by Maxquant 1.5 using default parameters.

#### 4.3.3. Data Analysis

Quantitative analysis of the entire hippocampus pre and post running wheel exposure was performed including only proteins that were detected in all samples (57 samples). An adjusted *p*-value < 0.20 was considered as statistically significant. To analyze subregion specific proteomic changes, log-transformed LFQ intensity values were first normalized using variance stabilizing transformation and missing intensity values were imputed using manual impute (shift: 1.8, scale 0.3) as provided by the ‘DEP’ bioconductor package (version 1.6.0). For GO term analysis, the String database (excluding text mining) with a default parameter [45,46].

### 4.4. Statistics

Data are presented as mean ± SEM. Statistics were calculated using the software GraphPad Prism (Version 5; GraphPad, San Diego, CA, USA). For quantification, all data were first tested for Gaussian distribution. Then, for the comparison of two groups Mann–Whitney U test, for the comparison of more than two groups Kruskal–Wallis test was used to determine the *p*-values. Grubb’s test (GraphPad) was used to determine outliers. *p* < 0.05 was considered statistically significant if not stated otherwise. For proteome analysis, a Student’s *t*-test was used.

## Figures and Tables

**Figure 1 ijms-21-03501-f001:**
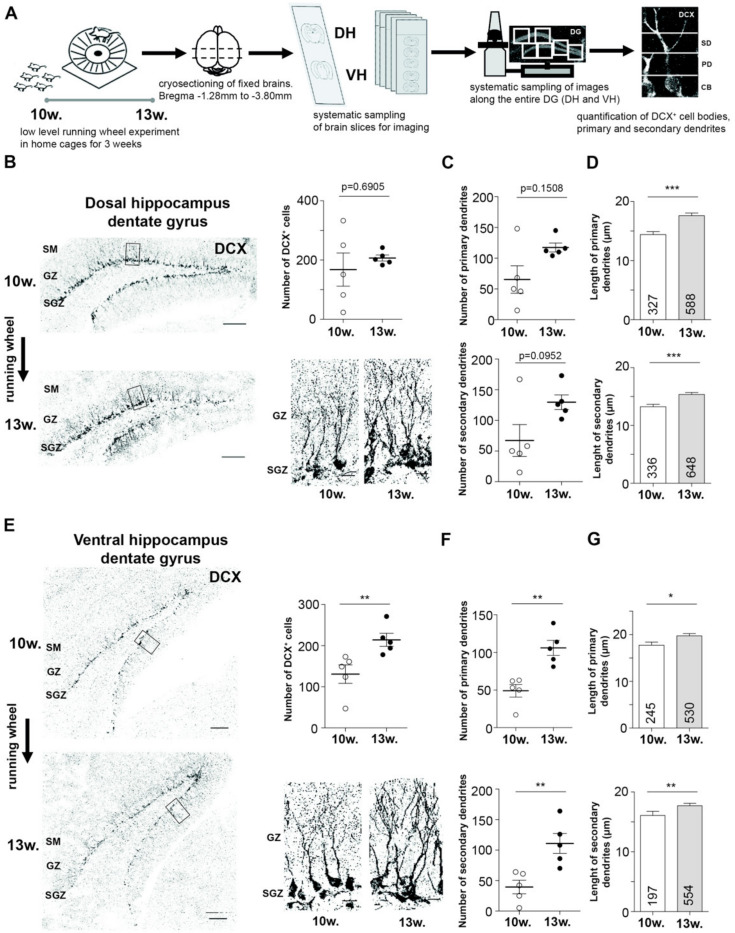
Running wheel exposure differently impacts adult neurogenesis in the dorsal versus ventral hippocampus. (**A**) Scheme showing the experimental strategy to enhance neurogenesis by running wheel exposure and survey differences between dorsal and ventral dentate gyrus (DG). (**B**,**E**) Representative immunohistochemical stainings (left) including insets (right) against doublecortin (DCX) for immature neurons prior and upon running wheel exposure (left). Quantification of DCX+ cell numbers is shown (right) for dorsal (B) and ventral (E) DG. (**C**,**F**) Quantification of number of primary and secondary dendrites of DCX+ neurons in dorsal (c) and ventral (F) DG. (**D**,**G**) Quantification of the mean length of primary and secondary dendrites of DCX+ cells in dorsal (D) and ventral (G) DG. Number of circles represent different animals, *n* = 5 for each group, * *p* < 0.05, ** *p* < 0.01, *** *p* < 0.001, mean ± SEM, Scale bar overview: 100 µm, scale bar inset: 20 µm.

**Figure 2 ijms-21-03501-f002:**
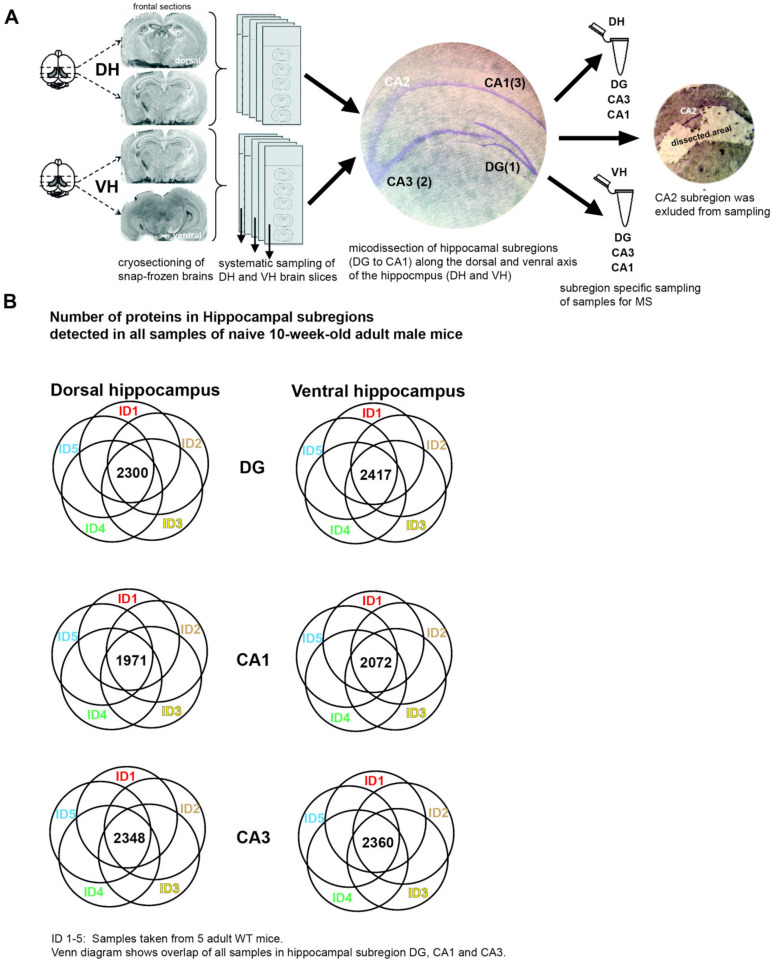
Label free quantitative mass spectrometry detects proteomic alterations during neurogenesis. (**A**) Experimental procedure of systematic sampling for dorsal and ventral hippocampal subregions. Subregions were manually microdissected. CA2 was excluded from analysis. (**B**) Commonly detected proteins in 5 different naïve animals (ID) for dorsal and ventral hippocampal subregions.

**Figure 3 ijms-21-03501-f003:**
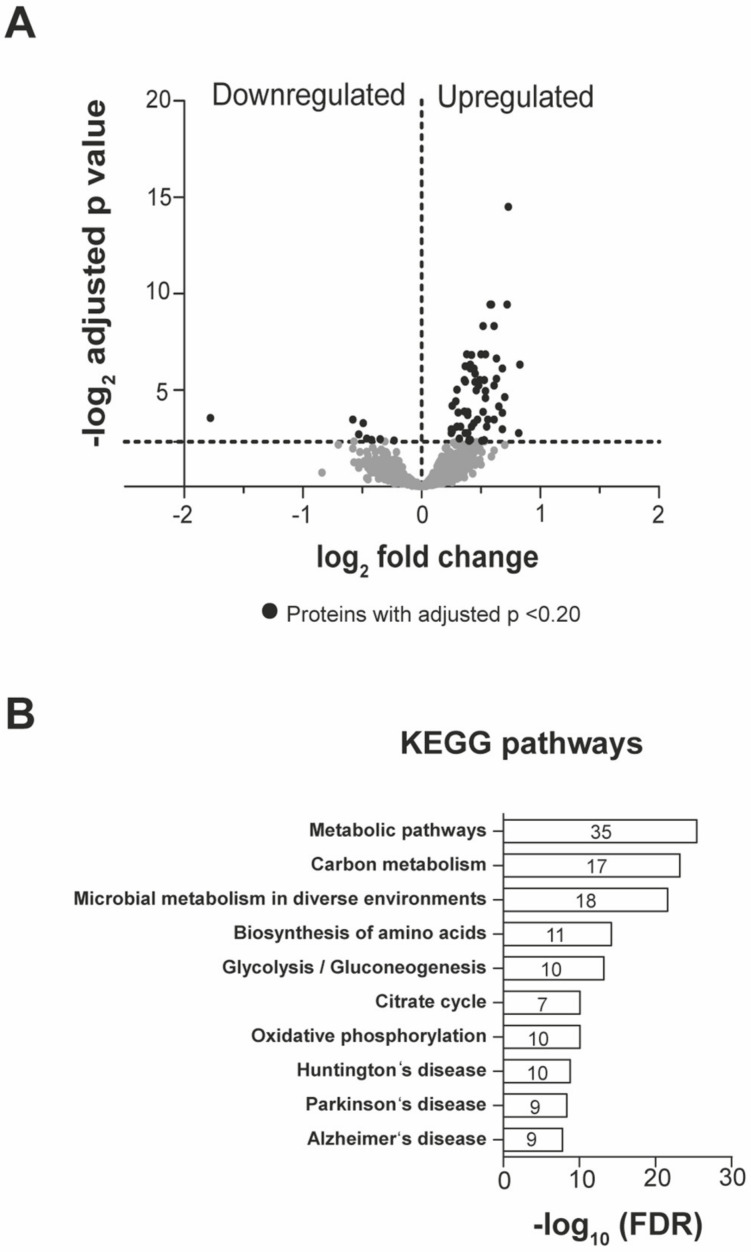
Voluntary exercise increases the expression of metabolic enzymes in the hippocampus. (**A**) Volcano plot of quantitatively detected proteins in all samples comparing running wheel exposed with naïve animals. Black dots represent significantly affected proteins (adjusted *p*-value < 0.2). (**B**) Functional clustering of proteins using the KEGG annotation. Data was analyzed using the String database.

**Figure 4 ijms-21-03501-f004:**
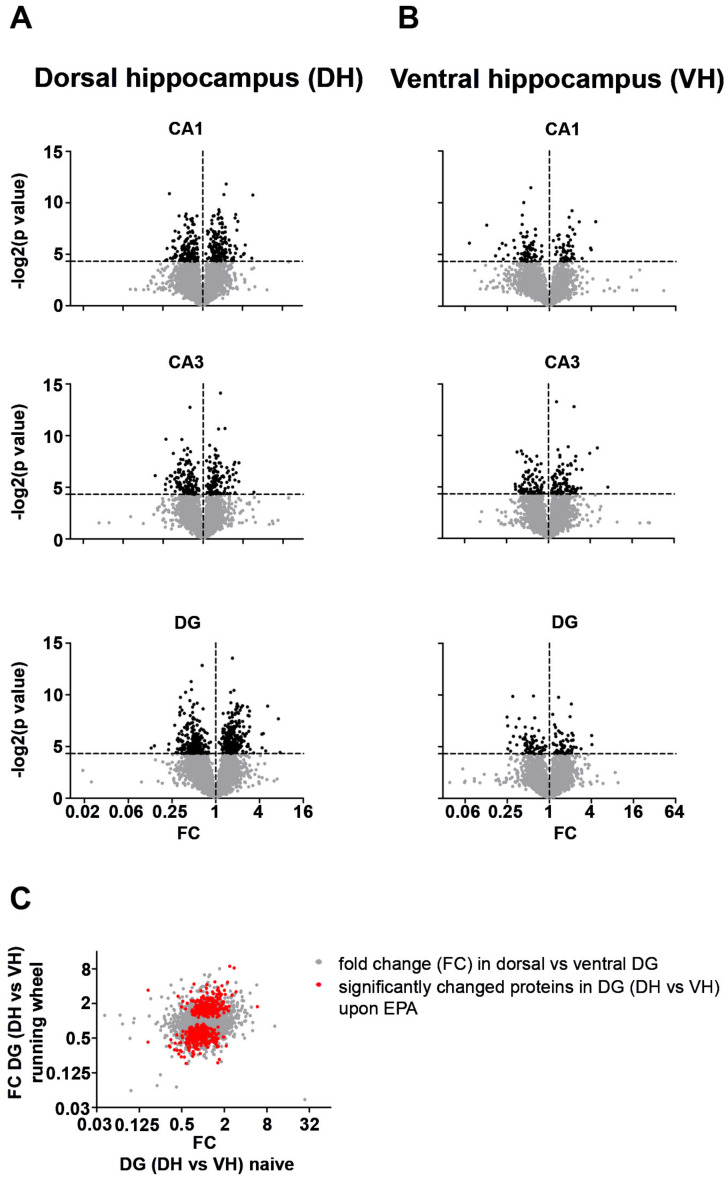
Dorsal and ventral hippocampal subregions show different proteomic adaptations in response to enhanced neurogenesis. (**A,B**) Volcano plots showing the fold change (FC) and *p*-value for dorsal (A) and ventral (B) hippocampal subregions. (**C**) FC for proteins in the dorsal versus ventral DG prior and upon running wheel exposure. Red dots indicate significantly affected proteins upon enhanced physical activity (EPA). Voluntary exercise increases the expression of metabolic enzymes in the hippocampus.

**Figure 5 ijms-21-03501-f005:**
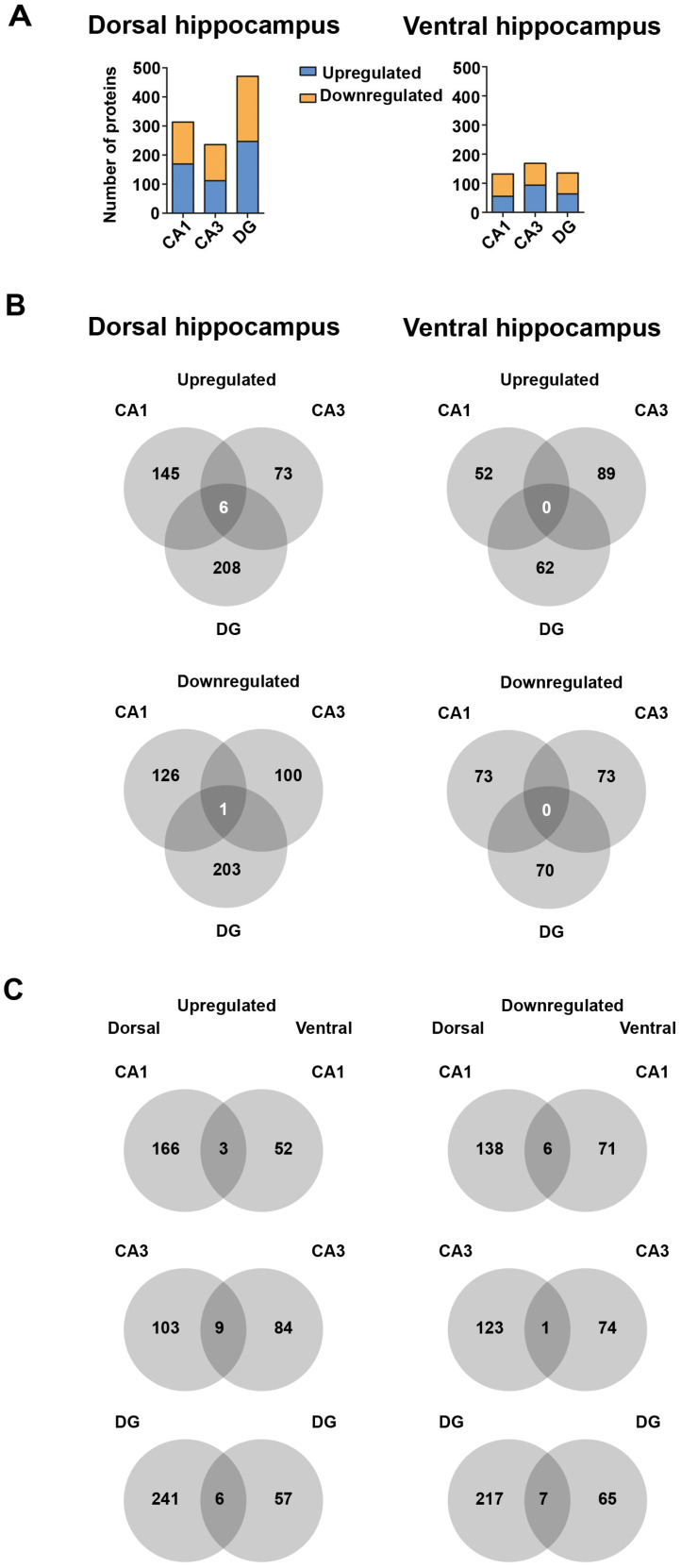
Hippocampal subregions are distinct in their proteomic alterations along the dorso-ventral axis. (**A**) Number of significantly upregulated and downregulated proteins for CA1, CA3 and DG of the dorsal (left) and ventral (right) hippocampus. (**B**,**C**) Venn diagrams comparing significantly affected proteins between hippocampal subregions (B) and between dorsal and ventral subregions (C).

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
