# Peer review of "Physical Activity Dynamically Regulates the Hippocampal Proteome along the Dorso-Ventral Axis"

_ijms, 2020, doi:10.3390/ijms21103501_

Round 1

Reviewer 1 Report

The article “Physical activity dynamically regulates the hippocampal proteome along the dorsal-ventral axis” attempts to look at the molecular underpinnings of enhanced physical activity (EPA) on increased hippocampal neurogenesis. This is an important topic, but the article comes across as quite vague in both its aim and its findings.

Below are some major concerns:

The terminology of the dorsal versus ventral hippocampal portions as distinct parts is misleading, especially considering that there is an anatomical gradient of afferent connectivity that likely also specifies a gradient of function. Futhermore, the authors talk about spatial navigation (as cognition) for the dorsal part of the hippocampus and state that the ventral part of the hippocampus is related to emotion. This is a vague description and the literature indicates that both portions of the hippocampus are involved in forming a cognitive map of space, although the ventral hippocampus may play a role in linking contexts to the emotion. It should be noted that this is not the full extent to which the hippocampus is involved in cognitive processing. If the authors are simply trying to distinguish between a dorsal and ventral portion in order to justify looking at differences along the axis, this justification could be done according to molecular gradients, anatomical gradients, and functional gradients without discretizing roles in function that are not actually that clear cut.

The authors talk about synaptic plasticity as well as neurogenesis, but what the authors consider to be differences or similarities between these two functions is not established. Such clarification would be helpful for understanding their goals and interpreting their results.

On page 2, paragraph 1, the authors say “Different pathways have been identified that are enhanced by EPA.” This is a fairly vague statement. Do the authors mean cellular or molecular pathways? Are they talking about something else? They go on to say that some of these pathways improve hippocampal neurogenesis, could these pathways be about pathways of information flow in the brain?

Results: The results seem promising but fall quite short of providing information.

  1. Although the authors noted that they do not see differences in naïve animals regarding neurogenesis levels between 10wk and 13wk animals, it is not clear at all why they used animals either 10 or 13wks and the conditions are confusing.
  2. Of note (not a concern): It is interesting that the authors find quantitative differences in neurogenesis along the dorsal-ventral axis in response to EPA.
  3. There are interesting differences hippocampus-wide of EPA versus naïve mice that extend the entire dorsal-ventral axis. These results include KEGG pathways that offer insight into functional changes induced by EPA.
  4. While the merit in the amount of overlapping differentially expressed proteins in the subregions of dorsal and ventral hippocampus are appreciated, why didn’t the authors assess enrichment of pathways as was done for the entire hippocampus? In general, I had a hard time following the logic behind interpretation (or speculation) of the results as presented. What do the authors gain from comparing the number of overlapping differentially expressed proteins in the subregions of the hippocampus if neurogenesis only occurs in the dentate gyrus? Are the authors suggesting that some kind of feedback within the hippocampus is contributing to hippocampal neurogenesis in the dentate gyrus? In particular, I didn’t understand the sentence on page 7, paragraph 1 “Together, it is tempting to speculate that DG, CA1 and CA3 exploit different expression programs adapt to EPA.” Should this sentence say “to adapt to EPA?
  5. Were corrections done for multiple comparisons? Why did the authors set a significance p-value of < 0.20, which seems rather high?

Discussion:

  1. The authors state that their findings of higher neurogenesis in the ventral hippocampus is in line with two independent studies, but then cite three papers. This brings into question if the histological studies are warranted or if they are redundant and thus used mice that might have otherwise not been needed (i.e. could the number of animals used in this study been reduced, or is there another justification for looking at exercise induced neurogenesis along the dorsal-ventral axis of the hippocampus?)
  2. It is unclear how reference 26 relates to the current study. This is particularly of interest given that the authors justify using proteomics to examine functional changes along the dorsal-ventral axis of the hippocampus in response to exercise by saying that transcriptome changes don’t correlate well to protein changes.
  3. (minor concern) The discussion is succinct, which is nice in some ways, but it could also be a bit more elegantly fleshed out.

Author Response

Reviewer 1

The terminology of the dorsal versus ventral hippocampal portions as distinct parts is misleading, especially considering that there is an anatomical gradient of afferent connectivity that likely also specifies a gradient of function. Furthermore, the authors talk about spatial navigation (as cognition) for the dorsal part of the hippocampus and state that the ventral part of the hippocampus is related to emotion. This is a vague description and the literature indicates that both portions of the hippocampus are involved in forming a cognitive map of space, although the ventral hippocampus may play a role in linking contexts to the emotion. It should be noted that this is not the full extent to which the hippocampus is involved in cognitive processing. If the authors are simply trying to distinguish between a dorsal and ventral portion in order to justify looking at differences along the axis, this justification could be done according to molecular gradients, anatomical gradients, and functional gradients without discretizing roles in function that are not actually that clear cut.

Authors response: Thank you for pointing this out. We revised the introduction stating that there is an anatomical gradient from dorsal to ventral hippocampus, which is likely to suggest functional differences. In addition, we also refer to transcriptome studies underlining the importance to investigate differences in the proteome along the dorso-ventral axis.

The authors talk about synaptic plasticity as well as neurogenesis, but what the authors consider to be differences or similarities between these two functions is not established. Such clarification would be helpful for understanding their goals and interpreting their results.

Authors response:We agree and modified this part in the introduction. We now refer to synaptic plasticity of neuronal circuits that allow the integration of newly born neurons. We hope that this finds the approval of this referee.

On page 2, paragraph 1, the authors say “Different pathways have been identified that are enhanced by EPA.” This is a fairly vague statement. Do the authors mean cellular or molecular pathways? Are they talking about something else? They go on to say that some of these pathways improve hippocampal neurogenesis, could these pathways be about pathways of information flow in the brain?

Authors response: We thank this referee for pointing this out. We changed this sentence in the introduction to ‘Different mechanisms and molecular pathways have been found to play a role in EPA enhanced cognition including cardiovascular and immunological effects.’ We tried to link the identified molecular signaling pathways such as IGF-1and BDNF with their impact on specific cellular pathways, e.g. neurogenesis. We also included the impact of neurogenesis on forgetting and cognitive flexibility.

Results:

The results seem promising but fall quite short of providing information.

Although the authors noted that they do not see differences in naïve animals regarding neurogenesis levels between 10wk and 13wk animals, it is not clear at all why they used animals either 10 or 13wks and the conditions are confusing.

Authors response: We agree. We clarified the usage of 10 week old animals and why we exposed them with a running wheel for 3 weeks in the revised results section.

 Of note (not a concern): It is interesting that the authors find quantitative differences in neurogenesis along the dorsal-ventral axis in response to EPA.

Authors response: We thank this referee for pointing this out.

There are interesting differences hippocampus-wide of EPA versus naïve mice that extend the entire dorsal-ventral axis. These results include KEGG pathways that offer insight into functional changes induced by EPA.

Authors response: Thank you for stressing this point.

 While the merit in the amount of overlapping differentially expressed proteins in the subregions of dorsal and ventral hippocampus are appreciated, why didn’t the authors assess enrichment of pathways as was done for the entire hippocampus? In general, I had a hard time following the logic behind interpretation (or speculation) of the results as presented. What do the authors gain from comparing the number of overlapping differentially expressed proteins in the subregions of the hippocampus if neurogenesis only occurs in the dentate gyrus? Are the authors suggesting that some kind of feedback within the hippocampus is contributing to hippocampal neurogenesis in the dentate gyrus? In particular, I didn’t understand the sentence on page 7, paragraph 1 “Together, it is tempting to speculate that DG, CA1 and CA3 exploit different expression programs adapt to EPA.” Should this sentence say “to adapt to EPA?

Authors response: Thank you for pointing this out. We performed gene ontology analysis to investigate potential clustering of biological processes for the different hippocampal subregions along the longitudinal axis. This data is now included in a new figure panel in the supplementary section (Supplementary Figure S3 - see attachment). Moreover, we explained why we included CA1 and CA3 in our analysis in the introduction and results section. Synaptic signaling in the hippocampus depends on a complex interaction between CA1, CA3 and DG. We reasoned that enhanced neurogenesis in the DG is followed by increased integration of newly born neurons upon 3 weeks. Through altered synaptic signaling, CA1 and CA3 might adapt to these changes. Moreover, we compared significantly changed proteins in CA1, CA3 and DG with each other to stress the point that EPA enhances expression differences between the subregions as shown for DG 

Were corrections done for multiple comparisons? Why did the authors set a significance p-value of < 0.20, which seems rather high?

Authors response: We thank this referee for the comment. We adjusted for multiple testing using the Benjamini-Hochberg procedure (FDR). The threshold < 0.2 applies to the FDR and not the raw p-value and indicates the maximum fraction of false positives in the result set. 20 % is not very conservative but within the range of commonly used cutoff values.

Discussion:

The authors state that their findings of higher neurogenesis in the ventral hippocampus is in line with two independent studies, but then cite three papers. This brings into question if the histological studies are warranted or if they are redundant and thus used mice that might have otherwise not been needed (i.e. could the number of animals used in this study been reduced, or is there another justification for looking at exercise induced neurogenesis along the dorsal-ventral axis of the hippocampus?)

Authors response: Thanks for the comment. For the neurogenesis in the ventral hippocampus we refer to two research articles and one review article. Since the review article is nicely summarizing differences in neurogenesis in ventral and dorsal hippocampus, we thought it would be appropriate to include it as well. However, we separated and subsequently specified the citations in the discussion part. Importantly, the two studies that showed differences in neurogenesis along the longitudinal axis did not use a running wheel to enhance neurogenesis. Moreover, they used mild stress to influence neurogenesis. Therefore, we cannot compare our model system with their data, which makes it necessary to investigate neurogenesis along the dorso-ventral axis under our conditions. Moreover, we used a higher number of animals because of our label-free mass spectrometry approach that critically depends on tissue and sample preparation.

It is unclear how reference 26 relates to the current study. This is particularly of interest given that the authors justify using proteomics to examine functional changes along the dorsal-ventral axis of the hippocampus in response to exercise by saying that transcriptome changes don’t correlate well to protein changes.

Authors response: We thank this referee for pointing this out. We included this reference to show that hippocampal gene expression is distinct. We also included expression studies along the longitudinal axis showing that there are different expression gradient in the hippocampus. The rational for referring to these studies is the lack of protein-wide studies addressing this important question.

(minor concern) The discussion is succinct, which is nice in some ways, but it could also be a bit more elegantly fleshed out.

Authors response: We agree. We revised and expanded the discussion part.

Reviewer 2 Report

The authors investigated the effects of the enhanced physical activity (EPA) on hippocampus, showing changes in the dendritic number and in protein expression with upregulation of 70 different proteins. Moreover, they revealed regional differences in response to EPA, noticing that dorsal hippocampal regions had higher number of affected proteins than ventral ones. The work is very interesting and well-written. There is only an item to improve: it concerns to the introduction section and regards to the description of previous evidences of EPA-induced changes on the hippocampal cell pathways.

Author Response

Reviewer 2

The authors investigated the effects of the enhanced physical activity (EPA) on hippocampus, showing changes in the dendritic number and in protein expression with upregulation of 70 different proteins. Moreover, they revealed regional differences in response to EPA, noticing that dorsal hippocampal regions had higher number of affected proteins than ventral ones. The work is very interesting and well-written.

There is only an item to improve: it concerns to the introduction section and regards to the description of previous evidences of EPA-induced changes on the hippocampal cell pathways.

Authors response: We thank this referee for pointing this out. We revised the introduction section and now summarized what is known about EPA-induced changes in hippocampal pathways.

Round 2

Reviewer 1 Report

The majority of the points were address in a satisfactory manner. However, upon reading the author responses and rereading the manuscript a major concern has come up. 

My previous review questioned the use of 10 week and 13 week old animals and ask for clarification. While the authors' response in the text was informative, I realized that my original question regarding this matter was unclear, for which I apologize. I had hoped that the authors would clarify in the text and within the figures that the animals without EPA were always 10 weeks old and that the animals with EPA were always 13 weeks old. I was confused the first round because the authors state that they did not see a difference in neurogenesis between 10 week and 13 week old animals without EPA, making me think that the original design included three groups of animals: 10 week old mice, 13 week old mice without EPA and 13 week old mice with EPA. This does not appear to be the case. 

The supplemental figure for the comparison of 10 week and 13 week old mice without EPA consists of only two mice per group. I find it difficult to believe that with only two mice per group that differences could be detected in either direction. Furthermore, there appears to be more variability in the dorsal hippocampus, where the proteome is supposedly more dynamic, and where differences in neurogenesis did not reach significance. Could this variability in neurogenesis at 10 weeks mask the differences that might occur between this group and the 13 week animals who had EPA.

There is a second major concern that arises for this study, namely, is there a difference between the proteomes of 10 week and 13 week old mice that did not have EPA? Because it appears that this was indeed the study design (which I was hoping for clarification about), the conclusions from this study depend on there not being a difference between these two ages. The appropriate design would have been to look at 13 week old mice without EPA and compare them to 13 week old mice with EPA in all aspects of the study. Supplemental Figure 1 "feels" like an afterthought due to inappropriate design, and while, had there been enough animals to state there really is not a difference in neurogenesis, the study would have had to prove there was also not a difference in proteome.

Thus, the authors state (Page 10, line 227): “ Therefore, our data further substantiate the previous notion that the hippocampus consists of distinct subdomains along the longitudinal axis.” This is actually only true if it is known that there is no difference in the proteomes of 10 week and 13 week mice without EPA. Otherwise, to justify this statement, the authors would have had to look at differential protein expression between the dorsal and ventral axis of 10 week old animals, which they could have done, but appear to not have done.

A few minor comments have also come up:

Page 2, lines 90-93 should be one sentence.

Page 2, line 92: “it has been shown reported” (needs edited).

Figure 2 legend, it is unclear as to why the last sentence is there about the running wheel differentially affecting neurogenesis in the dorsal and ventral hippocampus.

Page 10, line 220 suggests that distinct pathways (again, uncertain if this is about information flow/connectivity or molecular pathways) might influence neurogenesis in the dorsal and ventral hippocampus (which are again inappropriately referred to as two separate structures). The authors then reference a study that examines differential expression of mRNAs in CA2 of the hippocampus. The cited paper (reference 33) examined subregions of the dorsal hippocampus but did not look at dorsal vs. ventral hippocampus, thus, it is unclear how reference to this article supports the authors’ claim that different pathways may lead to differential neurogenesis in the dorsal and ventral portions of the hippocampus.

Author Response

Rebuttal letter Frey et al.

Reviewer: The majority of the points were address in a satisfactory manner. However, upon reading the author responses and rereading the manuscript a major concern has come up.

My previous review questioned the use of 10 week and 13 week old animals and ask for clarification. While the authors' response in the text was informative, I realized that my original question regarding this matter was unclear, for which I apologize. I had hoped that the authors would clarify in the text and within the figures that the animals without EPA were always 10 weeks old and that the animals with EPA were always 13 weeks old. I was confused the first round because the authors state that they did not see a difference in neurogenesis between 10 week and 13 week old animals without EPA, making me think that the original design included three groups of animals: 10 week old mice, 13 week old mice without EPA and 13 week old mice with EPA. This does not appear to be the case. 

The supplemental figure for the comparison of 10 week and 13 week old mice without EPA consists of only two mice per group. I find it difficult to believe that with only two mice per group that differences could be detected in either direction.

Authors response:We appreciate the reviewers concern. For the main focus of this study, we compared 5 naïve, 10 week old animals with 5, 13 week old, running wheel exposed mice. The number of animals we used for this part allowed us to detect significant differences between hippocampal subregions along the longitudinal axis. For our control experiment, we agree that the number of animals might not be sufficient to detect significant differences. However, due to the data distribution and calculated p-values, we are convinced that increasing the animal number would not reach any significance between 10 and 13 week old mice. In addition, when we designed the study, we aimed at reducing the number of animals used for this control condition for animal welfare reasons, a concern that was explicitly raised by this referee in the first round of review.

Reviewer: Furthermore, there appears to be more variability in the dorsal hippocampus, where the proteome is supposedly more dynamic, and where differences in neurogenesis did not reach significance. Could this variability in neurogenesis at 10 weeks mask the differences that might occur between this group and the 13 week animals who had EPA.

Authors response: We agree. Therefore, we included a sentence on the greater variance in the dorsal hippocampus of naïve, 10 week old mice (p. 3, l. 92-95). We would like this reviewer also to consider that this variability might contribute to the role of the dorsal hippocampus in spatial memory formation.

Reviewer: There is a second major concern that arises for this study, namely, is there a difference between the proteomes of 10 week and 13 week old mice that did not have EPA? Because it appears that this was indeed the study design (which I was hoping for clarification about), the conclusions from this study depend on there not being a difference between these two ages. The appropriate design would have been to look at 13 week old mice without EPA and compare them to 13 week old mice with EPA in all aspects of the study. Supplemental Figure 1 "feels" like an afterthought due to inappropriate design, and while, had there been enough animals to state there really is not a difference in neurogenesis, the study would have had to prove there was also not a difference in proteome.

Authors response: We thank this referee for this comment. The original design of this study consisted of a total of 2 (10 weeks naïve vs 10 + 3 weeks of running wheel exposure) experimental conditions with 5 mice per group, each. Therefore, the main focus of our current study was to investigate the influence of physical activity (EPA) on the hippocampal proteome within these 3 weeks. We compared 13 week old, running wheel exposed animals to 10 week old, naive mice for three main reasons: (i) our strategy mimics clinical conditions in which post- and pretreated conditions are compared. Particularly physical activity has been suggested as potential therapy to treat neurodegenerative diseases. (ii) according to our gene ontology analysis of hippocampal subregions (see Suppl. Figure 3), most of the affected pathways are related to neurogenesis. Therefore, we concluded that the generation of new neurons is the main driving force of EPA induced proteomic alterations in the hippocampus. Importantly, we did not observe differences in the number of doublecortin cells between 13 and 10 week old mice indicating that neurogenesis has not been affected within this time period. (iii) recent studies addressing proteomic changes during aging comparing 1 month or 6 month old mice with 12 or 24 month old animals, respectively, reveal minor changes in the proteome (~ 3-5%) (Duda et al. 2018, Ori et al. 2015) indicating that aging is not significantly altering the steady-state proteome of the mature hippocampus. Based on these results, we concluded that proteomic changes between 13 and 10 week old animals with running wheel exposure are minor. Supportive for this notion is a recent study showing that protein expression of 5 week and 20 week old animals is very similar (Gonzalez-Lozano et al. 2016). However, if this reviewer insists, we would compare published proteomics on aging with our datasets even though the experimental conditions are clearly different between the studies.

Reviewer: Thus, the authors state (Page 10, line 227): “Therefore, our data further substantiate the previous notion that the hippocampus consists of distinct subdomains along the longitudinal axis.” This is actually only true if it is known that there is no difference in the proteomes of 10 week and 13 week mice without EPA. Otherwise, to justify this statement, the authors would have had to look at differential protein expression between the dorsal and ventral axis of 10 week old animals, which they could have done, but appear to not have done.

Authors response: We thank this referee for pointing this out. According to our explanation (see above), we think that our analyses support the notion that hippocampal subdomains exhibit distinct proteomes. We see, however, the point of this referee and now rephrase our statement on page 10 to ‘Therefore, our data suggest that hippocampal subregions display significant differences within their proteomes along the longitudinal axis.’’

Reviewer: A few minor comments have also come up:

Page 2, lines 90-93 should be one sentence.

Page 2, line 92: “it has been shown reported” (needs edited).

Authors response: Thank you for pointing this out. We changed these sentences.

Reviewer: Figure 2 legend, it is unclear as to why the last sentence is there about the running wheel differentially affecting neurogenesis in the dorsal and ventral hippocampus.

Authors response: We thank this referee for the comment. We changed the legend of figure 2 accordingly.

Reviewer: Page 10, line 220 suggests that distinct pathways (again, uncertain if this is about information flow/connectivity or molecular pathways) might influence neurogenesis in the dorsal and ventral hippocampus (which are again inappropriately referred to as two separate structures). The authors then reference a study that examines differential expression of mRNAs in CA2 of the hippocampus. The cited paper (reference 33) examined subregions of the dorsal hippocampus but did not look at dorsal vs. ventral hippocampus, thus, it is unclear how reference to this article supports the authors’ claim that different pathways may lead to differential neurogenesis in the dorsal and ventral portions of the hippocampus.

Authors response: Thank you for pointing this out. We specified, which pathways might be regulated by EPA (‘Thus, distinct molecular pathways might convey the generation of neurons in these two structures.’, p. 10, l. 222-223). Moreover, we specified reference 33 in the text to emphasize gene expression differences in different hippocampal subregions (‘Furthermore, hippocampal subregions exhibit distinct mRNA expression patterns as shown for the CA2 subregion [33].’ ,p. 10, l. 224-225.

Round 3

Reviewer 1 Report

The response by the authors to my review questions is mostly satisfactory, but the responses to my points are not incorporated into the manuscript and I think the manuscript would benefit greatly by incorporating their logic. 

In particular, the response the authors gave about the stable proteome, with references, is clearly important to the interpretation of the results and citing these studies should be included in the manuscript to substantiate their study design. This justification, plus the justification of what their study was aim at completing, are well-intended and well-described and should be included in the manuscript. Including these points in the manuscript would certainly address my initial concern that the study was a vague and the question unclear. Clearly the authors have thought through their design and the manuscript would benefit greatly from describing this.

I still question the merit of not being able to detect differences based only on two animals per group, but given the other references they cite for the stable proteome, I think this point is not that important. 

It should be noted that the sentence for the variability of the cells in the dorsal HPC can be found in version 2 of the manuscript, but not version 3 and that the sentence that clarifies reference 33 being about subregion-specific changes in mRNA is in version 3, but not version 2.

Regarding the latter point about reference 33. I don't want to beat this into the ground, but only after our discussions have I realized where my confusion has come from, and the authors could address this for future readers. The authors site other papers that show subregion specific mRNAs, including differential expression along the longitudinal axis. Why point out CA2 specifically, especially when the authors reference an article specific to CA1? I think the logic is still a bit vague, ironically due to the specific mention of CA2. Why not just say there is subregion specific transcriptional differences along the dorsal-ventral axis (ref5, 6, and 33).

Finally, I will comment because the authors ask that I consider that the variability of neurogenesis in the dorsal hippocampus might contribute to the role of the dorsal hippocampus in spatial memory formation. Certainly variability in cell dynamics of any sort could potentially contribute to memory formation in general. What makes this variability in the dorsal hippocampus special for spatial memory formation? It seems like the type of information that comes into the dorsal hippocampus would be more likely to specify the type of memories formed. I recognize that it is likely the type of information in combination with changes in synaptic function (induced by input firing/neurogenesis/proteomic changes) that would dictate what memories are formed, but consider that we now know that the dorsal portion of the hippocampus is involved in creating and maintaining social memories, which perhaps have a spatial component to them, but which may not. Again, not important for this paper, but that being said, there is a lot of connectivity between the dorsal and ventral portions of the hippocampus. Overall, as a philosophical question for me to consider, I’m don’t understand what point the authors are trying to make.

Author Response

Reviewer: The response by the authors to my review questions is mostly satisfactory, but the responses to my points are not incorporated into the manuscript and I think the manuscript would benefit greatly by incorporating their logic. 

In particular, the response the authors gave about the stable proteome, with references, is clearly important to the interpretation of the results and citing these studies should be included in the manuscript to substantiate their study design. This justification, plus the justification of what their study was aim at completing, are well-intended and well-described and should be included in the manuscript. Including these points in the manuscript would certainly address my initial concern that the study was a vague and the question unclear. Clearly the authors have thought through their design and the manuscript would benefit greatly from describing this.

I still question the merit of not being able to detect differences based only on two animals per group, but given the other references they cite for the stable proteome, I think this point is not that important.

Authors response: Of course, we agree and included our clarification in the results part (p. 2-3, l. 89-98).

Reviewer: It should be noted that the sentence for the variability of the cells in the dorsal HPC can be found in version 2 of the manuscript, but not version 3 and that the sentence that clarifies reference 33 being about subregion-specific changes in mRNA is in version 3, but not version 2.

Authors response: Thank you for pointing this out. We checked the current version of the manuscript to ensure that all changes are now included. The sentence on the variability is now included (p. 3, l. 107/111.). We also rephrase the sentence that refers to reference 33 (current version reference 36)

Reviewer: Regarding the latter point about reference 33. I don't want to beat this into the ground, but only after our discussions have I realized where my confusion has come from, and the authors could address this for future readers. The authors site other papers that show subregion specific mRNAs, including differential expression along the longitudinal axis. Why point out CA2 specifically, especially when the authors reference an article specific to CA1? I think the logic is still a bit vague, ironically due to the specific mention of CA2. Why not just say there is subregion specific transcriptional differences along the dorsal-ventral axis (ref5, 6, and 33).

Authors response: We are now summarizing the papers on transcriptional differences in hippocampal subregions with the following sentence as suggested by this referee (‘Moreover, additional studies showed that there is subregion specific transcriptional differences along the longitudinal axis [5,6,36]’, p. 11, l. 233/234).